# Early Skin-to-Skin Contact in Preterm Infants: Is It Safe? An Italian Experience

**DOI:** 10.3390/children10030570

**Published:** 2023-03-17

**Authors:** Luca Bedetti, Licia Lugli, Natascia Bertoncelli, Eugenio Spaggiari, Elisabetta Garetti, Laura Lucaccioni, Federica Cipolli, Alberto Berardi

**Affiliations:** 1Neonatal Intensive Care Unit, Department of Medical and Surgical Sciences of Mothers, Children and Adults, University Hospital of Modena, 41124 Modena, Italy; 2PhD Program in Clinical and Experimental Medicine, University of Modena and Reggio Emilia, 41124 Modena, Italy; 3Pediatric Unit, Department of Medical and Surgical Sciences of Mothers, Children and Adults, University of Modena and Reggio Emilia, 41124 Modena, Italy

**Keywords:** skin-to-skin contact, kangaroo care, preterm infant

## Abstract

Background: Skin-to-skin contact (SSC) is one of the four components of kangaroo care (KC) and is also a valued alternative to incubators in low-income countries. SSC has also become a standard of care in high-income countries because of its short- and long-term benefits and its positive effect on infant growth and neurodevelopmental outcome. However, barriers in the implementation of SSC, especially with preterm infants, are common in NICUs because parents and health care professionals can perceive it as potentially risky for the clinical stability of preterm infants. Previous studies have assessed safety before and during SSC by monitoring vital parameters during short-time intervals. Aims: To demonstrate the safety of early SSC in preterm infants during at least 90 min intervals. Design: Prospective observational monocentric study. Methods: Preterm infants born between June 2018 and June 2020 with a gestational age of ≤33 weeks and a birth weight of <2000 g were monitored while performing an SSC session during the first three weeks of life. Infants with necrotizing enterocolitis, sepsis, and congenital malformations on mechanical ventilation or with more than five apneas in the hour before SSC were excluded. Continuous oxygen saturation (SaO2), heart rate (HR), and respiratory rate (RR) were registered during an SSC session and in the hour before. The minimum duration of an SSC session was 90 min. Information regarding postmenstrual age (PMA), body weight, respiratory support, presence of a central venous catheter and the onset of sepsis within 72 h after a session was collected. Two physicians, blinded to infant conditions and the period of analysis (before or during SSC), evaluated desaturation episodes (SaO2 < 85%, >15 s), bradycardia (HR < 100, >15 s) and apneas (pause in breathing > 20 s associated with desaturation and/or bradycardia). A Wilcoxon rank sum test was used for the statistical analysis. Results: In total, 83 episodes of SSC were analyzed for a total of 38 infants. The mean gestational age at birth was 29 weeks (range 23–33 weeks). Median PMA, days of life, and body weight at SSC were 31 weeks (range 25–34 weeks), 10 days (range 1–20 days), and 1131 g (range 631–2206 g), respectively. We found that 77% of infants were on respiratory support and 47% of them had a central venous catheter (umbilical or peripherally inserted central catheter) during SSC. The total duration of desaturation, bradycardia, and the number of apneas were not statistically different during the SSC session and the hour before. No catheter dislocation or ruptures were reported. Conclusions: These findings highlighted the safety of early SSC in preterm infants and the possibility of performing it in an intensive care setting in the first weeks of life. In addition, these findings should reassure health care professionals offering this practice as a standard of care. SSC plays a key role in the care of preterm infants due to its short- and long-term positive benefits, and it deserves to be increasingly offered to infants and their parents.

## 1. Introduction

Sensory stimulation (pain, noise, and light), tactile, and proprioceptive stimulation caused by daily routine care in the Neonatal Intensive Care Unit (NICU) is an early life stress exposure, which may have an adverse impact on the neurological development of preterm infants. Neurodevelopmental care in the NICU is performed through family-centered care, which became the care model for adapting the atypical environment of the NICU to the strengths and vulnerabilities of preterm infants. One of the core principles of family-centered care is kangaroo care (KC), which is a comprehensive term used for skin-to-skin contact (SSC), early and nearly exclusive breastfeeding, early discharge from hospital, and follow-up after discharge [1]. Compared to conventional care, SSC and KC have generally been shown to have benefits for the global health and development of preterm infants, and in high-income countries, SSC has become a standard of care [2,3,4,5] for preterm infants.

SSC influences the behavioral states of preterm infants and improves their organization in preterm infants. Indeed, SSC enhances the states of deep sleep and quiet wakefulness and decreases light sleep and drowsiness [6]. Moreover, KC and, in particular, SSC, have a positive effect on infant weight gain. A literature search including a systematic review, randomized controlled trials and non-randomized controlled trials demonstrated that KC is correlated with improved weight gain or reduced body weight loss among preterm infants [7]. SSC has been proven to have benefits for cortical activation and oxygen perfusion in preterm infants admitted to the NICU. Bembich et al. (2023) performed functional near-infrared spectroscopy monitoring of cortical activity during KC in preterm infants with a gestational age at birth below 32 weeks. Near-infrared spectroscopy monitoring was performed during the first 30 min of KC in the room where the infant was cared for, and the environment was kept as quiet as possible. After 30 min of KC, an adequate functional hemodynamic response and cerebral activation in the right motor and primary somatosensory cortex were observed. As a consequence, KC may be considered a neuromodulatory intervention in preterm infants [8].

Most studies focus on aspects of infant health when receiving SSC, but the effects of SSC on a mother’s health and attachment status are also important. A quasi-experimental study demonstrated that the rate of maternal attachment was significantly higher in mothers who experienced skin-to-skin contact compared to the rate of mothers who received traditional care. SSC practice had a positive impact on mothers’ mental health and on their relationship with their preterm infants [9].

Although SSC can be considered a neurodevelopmental intervention for preterm infants in the NICU, there are factors that may negatively influence parents, medical, and nursing staff in relation to SSC practice [10]. Barriers and facilitators to SSC implementation in the NICU were explored and identified in an umbrella review, and multiple different factors were identified, including professional, parent/family access, and cultural factors [11]. In a survey carried out by the Italian Neonatology Society on the implementation of KC in Italian NICUs, restrictive parents accessing policies were found to be the main barrier to the implementation of KC [12]. Moreover, a systematic review on barriers and enablers to KC implementation demonstrated that the main obstacle for nurses concerned infants’ clinical condition, extra workload, and a lack of guidelines and training. From the mothers’ point of view, fear of hurting their infants was the main obstacle to practicing KC [13]. The safety and effectiveness of SSC is a cornerstone to implementing its practice [14]. Safety implies the physiological stability of vital functions for preterm infants experiencing early SSC. Few studies define the physiological stability of vital functions that makes infants eligible for early SSC. In their qualitative analysis, Lee et al. (2012) reported significant variability and disagreement over the definition of clinical stability among different NICU professionals. According to their findings, age, body weight, respiratory distress, blood pressure, temperature, apnea, bradycardia, and desaturation events were some of the variables used to determine clinical stability, but there was a great variation in the criteria and specific parameters used [15]. In addition, a study on policies and practices regarding parental holding of infants in the U.S. neonatal units highlighted that the main concern regarding SSC was infant safety [16]. Bergman et al. (2004) stated that it is safe to start SSC within the first 6 h after birth in infants with a birth weight between 1200 and 2199 g. SSC reduces cardiorespiratory instability and the “hyper-arousal and dissociation” patterns described in infants in response to separation from their mothers [17]. Lorenz et al. (2017) measured and compared regional cerebral O2 (rcO2), oxygen saturation, heart rate, and inspired oxygen before, during, and after SSC in preterm infants in relation to respiratory support; their findings demonstrated that rcO2 is stable during SSC in very preterm infants with respiratory support, and SSC provides comparable physiological stability to incubator care in preterm infants with respiratory support [18]. Clinical stability and safety could therefore be key factors when providing SSC to preterm infants [19,20].

In our NICU, SSC is a common practice. This observational study aimed to assess whether changes in clinical stability occurred after starting SSC in order to reassure medical and nursing staff as to the safety of SSC practice.

## 2. Materials and Methods

### Study Design

This monocentric observational study was conducted between 1 June 2018 and 31 May 2020 at the Modena University Hospital, a high-volume level-3 NICU with 20 beds and 6 rooms. Approximately 3000 deliveries per year (including of approximately 60 very low birth weight and 30 extremely low birth weight infants) occur in the center. 

SSC is routinely performed in our unit and all nurses have been trained to guide and assist parents during the procedure. All bedrooms are equipped with an armchair for the parents beside each incubator. 

During the SSC, infants wear only a diaper and are placed prone on the parent’s naked chest, sternum to sternum; their heads are covered with a wool cap, they are in a slight sniffing position and their backs are covered by a blanket or the parent’s shirt (Figure 1).

Infants who underwent SSC during the first three weeks of life, who were born between 27 + 0 and 32 + 6 weeks’ gestation and with a birth weight under 2000 g were eligible for the study (Table 1). Parents were informed and consented to the SSC; the minimum expected duration of each SSC session was 90 min. Exclusion criteria were ongoing proven or suspected sepsis or necrotizing enterocolitis, the presence of congenital malformations, intraventricular hemorrhage (IVH) grade 3–4, ≥5 central apneas in the hour before an SSC session, and severely ill infants.

Continuous oxygen saturation (SpO2), heart rate (HR), and respiratory rate (RR) were recorded (Monitors IntelliVue MP40 Neonatal—Philips, Amsterdam, The Netherlands) during SSC sessions and in the hour before starting. Heart rate and oxygen saturation were recorded every 10 min, and we compared median values before and during an SSC session. 

During an SSC session, any activity with a potential impact on vital signs, such as crying, feeding, or suctioning, was recorded. Information regarding gestational age, body weight, respiratory support, presence of central venous catheter at the time of the SSC, and the onset of sepsis was collected. 

Once recordings were performed, two physicians, blinded to infant characteristics (i.e., gestational age, respiratory support) and the period of vital sign recording (prior or during SSC), evaluated the presence of desaturation (SpO2 < 85%, longer than 15 s), bradycardia (HR < 100, lasting more than >15 s) and apnea (pause in breathing lasting more than 20 s associated with desaturation and/or bradycardia). This study was approved by the Local Ethical Committee (protocol 559/2018) and all parents gave informed consent.

## 3. Statistical Analysis

A statistical analysis was performed by Stata/SE 13.0 (StataCorp LP). Sample characteristics were described using frequencies (percentages) for categorical variables and medians and interquartile ranges for continuous variables. The number and duration of the events (apnea, desaturation, and bradycardia) were proportionally calculated to sixty minutes, and an analysis of hourly events was performed. 

The Wilcoxon rank sum test was used to compare the number and duration of desaturation, bradycardia, and apnea before and during SSC. A *p*-value of <0.05 was the cut-off for statistical significance. A Mann–Whitney U test was used to compare heart rate and oxygen saturation values before and during an SSC session in the study population.

## 4. Results

### 4.1. Patients and SSC Session Characteristics

Over a period of 48 months, 38 infants (18 girls and 20 boys) were enrolled for a total of 83 SSC sessions. The median gestational age at birth among the study population was 29 weeks (range 23–33) and the median birth weight was 1099 g (range 540–1996). A total of 83 session of SSC were recorded. The median gestational age at the time of the first SSC session was 31 weeks (range 25–34), the median body weight at the time of the first SSC was 1130 g (range 630–2206), and the median days of life at the first SSC session was 10 days (range 1–20) (Table 1). Moreover, 1 SSC session was experienced by a mechanically ventilated infant, while the remaining 82 SSC sessions were performed in non-mechanically ventilated infants (non-invasive ventilation, *n* = 63; breathing room air, *n* = 19). A total of 66 out of 83 (79.5%) SSC sessions were performed in infants with a body weight under 1500 g. In 92.1% of cases, the SSC session was performed by mothers; in the remaining 7.9% of cases it was performed by fathers. Almost 80% of infants were on respiratory support and more than half of them had a central venous catheter in place during the SSC sessions (Table 1).

### 4.2. SSC Sessions and Vital Sign Stability

There was no difference between oxygen saturation and heart rate before and during an SSC session (Figure 2 and Figure 3).

There were no significant differences before and during an SSC session in relation to the number and duration of episodes of desaturation, bradycardia, and apnea (Table 2). 

### 4.3. Further Adverse Events

No displacement or rupture of central venous lines was reported after an SSC session. No further events (i.e., feeding tube dislocation) occurred during an SSC session. One case of late-onset sepsis occurred within 72 h of an SSC session.

## 5. Discussion

SSC is a non-pharmacological intervention and can be considered a nurturing touch. It has been proven that SSC modulates the effect of stress on autonomic nervous system activity, decreasing the “fight or flight” response and favoring the “rest and digest” one [21]. A prolonged period of stress, as often experienced by preterm infants in NICUs, can increase the activity of the autonomic nervous system (heart rate and heart rate variability ratios) and influence the short- and long-term neurodevelopmental outcome. Reducing preterm infants’ stress to minimum levels can promote neuroplasticity and make infants capable of adequately reacting to stress exposure. Skin-to-skin contact is a nurturing touch offered by parents to their infants, and it is the most effective strategy for reducing infant stress [22,23]. A mother’s skin-to-skin contact with her preterm infant provides multisensory stimulation in a unique and interactive way, including tactile, proprioceptive, vestibular, olfactory, auditory, visual, thermal, and emotional stimulation. SSC can positively affect and increase the maturation of the organization of behavioral states [6,24]. In addition, providing SSC to mothers of preterm infants in the NICU can be beneficial for mothers’ mental health. It can reduce the separation of the mother–infant dyad and promote an ongoing emotional connection. As a consequence, attachment between the mother–infant dyad will be facilitated, and the general wellbeing of infants and their parents will be improved [9]. Early SSC is also a nurture intervention based on the so-called “nurturescience”. Nurturescience is now considered an approach embedded in neuroscience. In nurturescience, emotional connection and relationship are the main and immediate developmental outcomes, while in neuroscience both emotional and relationship factors are considered as “secondary” developmental outcomes related to socioemotional development. Nurturescience is founded on the basic needs of a newborn and their parents. The key message of neuroscience is zero separation in the mother–infant dyad and the increase in immediate SSC. The nurturescience approach is therefore extremely important for preterm infants admitted to the NICU, because it promotes the implementation of immediate and early SSC, and can potentially reduce the risks of neurodevelopmental problems [24,25,26].

A recent Italian survey showed that despite the 24 h access in Italian NICUs almost doubling in the last 20 years and SSC becoming a routine practice, its frequency and duration are still insufficient when compared to WHO standards [27]. Therefore, providing safety data on this practice reinforces the belief in health care professionals that it should be performed whenever possible to improve the neurodevelopmental outcome of preterm infants. This study evaluates early SSC sessions performed in preterm infants during the first three weeks of life in an Italian NICU. In this study, the majority (80%) of infants who experienced an early SSC session were on ventilatory support, had a body weight of <1500 g, and had a central venous catheter in place. Our findings demonstrate the safety of SSC because there were no differences in heart rate, respiratory rate, or duration of desaturation before and during an SSC session. Furthermore, no catheter dislocation or ruptures were reported. These findings are in line with those of Carbasse et al. (2013), who investigated safety (i.e., absence of adverse events such as accidental extubation or worsened clinical condition) and physiological stability during an SSC session in 96 very low birth weight preterm infants in a NICU. SSC was offered whenever the clinician and nurse assessed the infants’ clinical stability. They found that SSC can promote physiological stability in this population of infants, leading to increased oxygen saturation, alongside a clinically significant decrease in supplemental oxygen requirements and greater stability in heart rate trends [28]. Our findings are consistent with those of Park et al. (2014), who performed a prospective study of 31 preterm infants from 25 to 32 weeks postmenstrual age (PMA). They started SSC at a mean age of 24.5 days in the 25–28 week PMA group and 11.9 days in the 29–32 week PMA group. They found no changes in vital signs before, during, and after an SSC session [29]. However, they used a very short time interval to evaluate potential changes (15 min before, 30 min during, and 15 min after the session). Our findings are also consistent with a meta-analysis comparing the physiological parameters of infants before, during, and/or after SSC in before and after studies, randomized controlled trials, and cohort studies. There was no evidence of differences in heart rate between, before, and during the SSC along with an increase in body temperature; a decrease in oxygen saturation was reported when SSC was offered in a colder environment [30]. 

Furthermore, temperature control is an additional and hugely important parameter in the care of fragile preterm infants in the NICU. Temperature control during immediate SSC (first postnatal hours) was checked by Linner A et al. (2020) in a randomized control study of 55 preterm infants with a gestational age under 33 weeks PMA and a birth weight under 2500 g. In their study, they found that SSC can be started in the first hour after birth. However, they stated that maintaining a stable temperature in room air may be a challenge when the parent is the only source of heat [31]. Maastrup R et al. (2010) investigated the maintenance of adequate temperature (36.5–37.5 °C) and an infant’s stability during the transfer and position to SSC. According to their findings, clinically stable, extremely preterm infants can maintain adequate thermal skin control and adequate stability during SSC when performed with their parents [32].

The safety of early SSC helps NICU professionals, nurses, and doctors increase its implementation in NICUs. Nurses who are skilled enough in SSC practice and have positive feelings about it are keener to offer and support it for parents and infants in the NICU. In contrast, those who are less confident with SSC may have misconceptions and uncertainties as to its suitability and safety. In their descriptive cross-sectional survey, Shattnawi et al. (2018) assessed the knowledge and perceptions of Jordanian NICU nurses in relation to KC. They found that the majority of nurses believed that KC has important and positive benefits for both the mother and infant, and they agreed that KC would improve the global health of preterm babies. Despite these positive perceptions, concerns still remain among nurses regarding the feasibility and safety of KC for infants who are intubated and have central catheters in situ [33]. Nation et al. (2021) designed a qualitative improvement project to increase SSC practice among infants born before 29 weeks’ gestation, which was independent from respiratory support. They performed a pre/post survey among nurses, which evaluated perceived barriers to SSC and comfort levels in offering SSC to preterm infants. The improvement project consisted an updated specific SSC NICU protocol and tailored education and training of NICU nurses. After the implementation of the qualitative project, SSC rates increased by up to 39.1%, and nursing comfort levels increased when SSC was performed in intubated infants and infants with a central catheter or umbilical venous catheter [34].

Since preterm infants are those at the greatest risk of infection, some may be concerned that SSC could be an additional risk factor. However, to date, studies report that SSC is associated with a reduction rather than an increase in sepsis in premature infants. In addition to a Cochrane review showing that KC programs significantly reduced the risk of serious infections/sepsis, a very recent study investigated the effects of SSC in the largest population of infants that have ever been evaluated in high-income countries [35,36]. The French prospective, nationwide, population-based study of very preterm infants EPIPAGE-2, evaluated 919 infants exposed to early SSC (SSC performed before day 4 of life). It found that SSC was not associated to late-onset sepsis or late-onset neonatal infection. The mechanisms underlying the potential protective role against the infection of SSC, seems to be related to changes in bacterial flora. Indeed, SSC modifies the microbiome and promotes skin colonization of antibiotic-sensitive bacteria [37]. This potential of SSC further underscores the neuroprotective role of SSC, given that infections are a crucial risk factor for long-term neurodevelopmental impairment [38]. Unfortunately, the design of our study did not allow us to make assessments on this issue. 

This study has several limitations. Firstly, we did not assess the stability of skin temperature during an SSC session since it was not strictly and regularly measured. Furthermore, the relatively small number of SSC sessions could have potentially underestimated differences in vital signs before and during an SSC session. Finally, only one neonate was mechanically ventilated during the SSC session, preventing us from obtaining information about the safety of SSC during mechanical ventilation.

## 6. Conclusions

SSC that is offered early is safe for infants, parents, and NICU staff. Parents are not only able to understand SSC, but also enjoy it. The more they perceive its safety, the more they are reassured to practice it; parents should provide nurturing touch, namely SSC, whenever possible to their preterm infant in the NICU. From a clinician’s point of view, early and safe SSC encourages nurses and doctors to offer it to parents of preterm infants. Nursing and medical comfort surrounding SSC is a key aspect to increasing SSC rates in the NICU.

SSC must become a standard of care for preterm infants admitted to the NICU and their parents. Further studies could assess the potential beneficial effects of SSC, even in less stable preterm infants in detail, which is an aspect not investigated in this study.

## Figures and Tables

**Figure 1 children-10-00570-f001:**
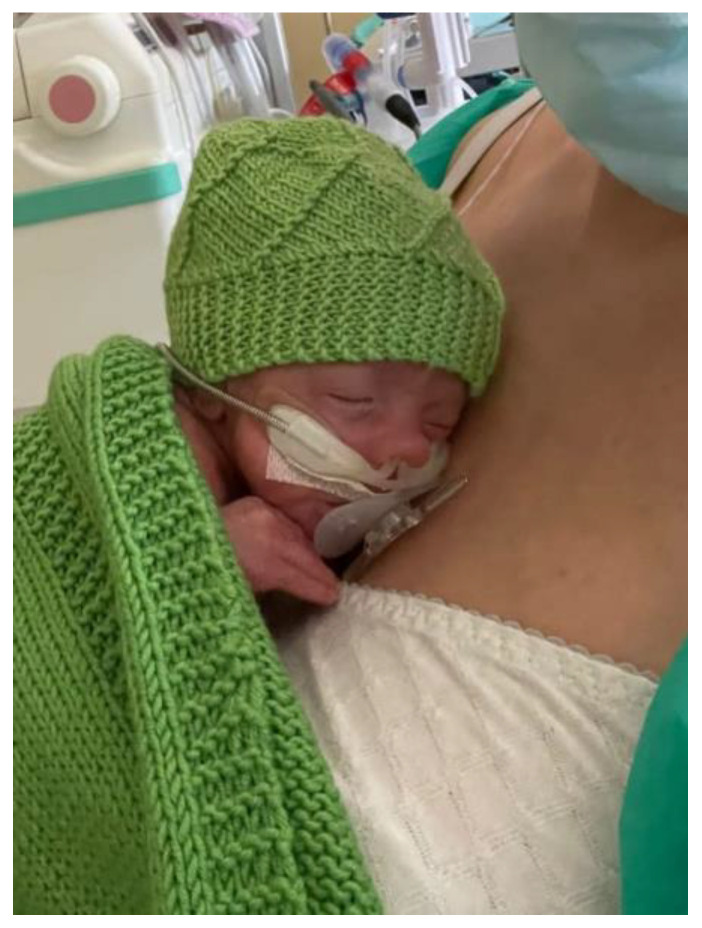
Mother–infant dyad during an SSC session.

**Figure 2 children-10-00570-f002:**
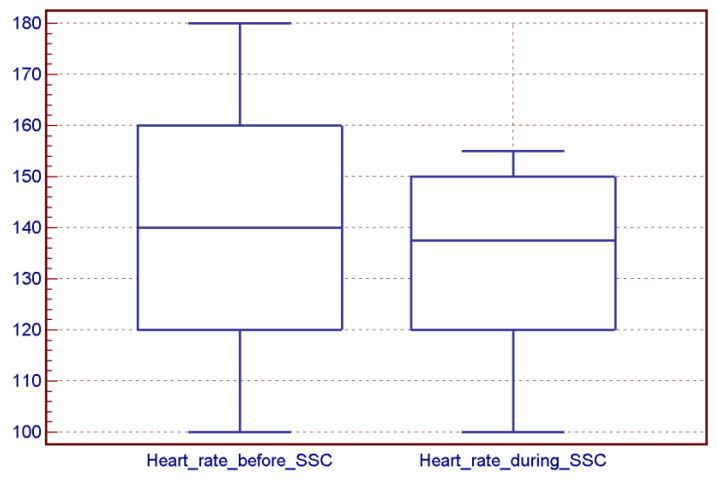
Infants’ heart rate before (median 140 beats per minute, range 100–180) and during an SSC session (median 137 beats per minute, range 100–155) (*p* = 0.22).

**Figure 3 children-10-00570-f003:**
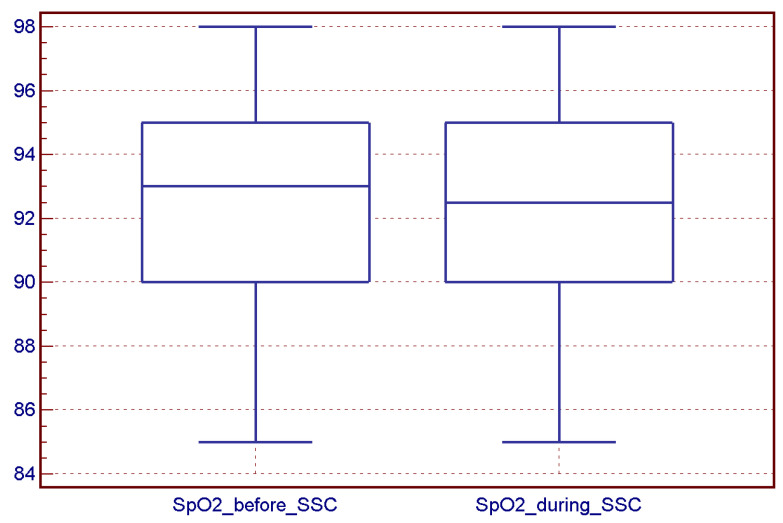
Infants’ SpO2 before (median 93.0, range 85–98) and during an SSC session (median 92.5, range 85–98) (*p* = 0.76).

**Table 1 children-10-00570-t001:** Clinical characteristics of infants before and during an SSC session.

	Results
Median gestational age, at birth, weeks, (range)	29 (23–33)
Median gestational age at SSC, weeks, (range)	31 (25–34)
Median birth weight, g, (range)	1099 (540–1996)
Median body weight at SSC, g, (range)	1130 (630–2206)
Median days of life at SSC (range)	10 (1–20)
Respiratory Support, *n* (%)	
− None	19 (22.9%)
− HFNC	6 (7.2%)
− NIV	57 (68.7%)
− Mechanical ventilation	1 (1.2%)
Catheters, *n* (%)	
− None	22 (26.5%)
− Peripheral venous catheter	14 (16.9%)
− Central venous catheter	47 (56.6%)

HFNC, high flow nasal cannula; NIV, non-invasive ventilation; SSC, skin-to-skin contact.

**Table 2 children-10-00570-t002:** Number and duration of apneas, desaturations, and bradycardias during and before an SSC session.

	Before an SSC Session (*n* = 83)	During an SSC Session (*n* = 83)	*p*
Sessions with apnea, *n* (%)	3 (2.7%)	6 (4.2%)	0.351
Sessions with desaturation, *n* (%)	12 (19.5%)	20 (30.6%)	0.314
Sessions with bradycardia, *n* (%)	10 (9%)	11 (13.3%)	0.481
Duration of desaturations, seconds (mean ± ds)	15–350 125 ± 118	15–560 115 ± 171	0.598
Duration of bradycardia, seconds (mean ± ds)	15–60 28 ± 15	15–135 36 ± 34	0.746

## Data Availability

The original contributions presented in the study are included in the article; further inquiries can be directed to the corresponding author/s.

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
