# Peer review of "Early Skin-to-Skin Contact in Preterm Infants: Is It Safe? An Italian Experience"

_children, 2023, doi:10.3390/children10030570_

Round 1

Reviewer 1 Report

This is a good paper and the manuscript is clearly written. The manuscript is good for publication. 

Author Response

Dear Reviewer

thank you for you comment and support to the publication of our article. We strongly believe that early SSC and kangaroo care in general can improve the global health and the quality of life of preterm infants and their parents.

Reviewer 2 Report

Dear Authors,

thank you for your work aiming to improve patient care in the NICU. The study presents very important data on the safety of early skin-to-skin contact in preterm infants. Despite the relatively small sample size of patients, the results support a crucial point, namely that SSC is safe and that medical professionals can feel confident in recommending it as a standard of care for infants and their parents. I believe that this was a well-designed study with reliable results. 

Author Response

Dear Reviewer,

thank you for your supportive comment. We strongly believe that the earlier the SSC is offered to infants and parents the better are the short and long term benefits in terms of relationship among the triad, neuro-developmental outcome and mental health of parents. We agree that SSC must become a standard of care in the NICUs worldwide.

We will improve the English language of our manuscript, as you suggest.

Reviewer 3 Report

1.       Aim is not requited in materials and methods section.

2.       Line 99-100: the said period was not affected due to covid 19.

3.       Format the table properly as per journal guidelines.

4.       Check the table 2 values again for correctness.

5.       Improve the language.

6.       Avoid long sentences.

Author Response

Dear Reviewer,

thank you for your comments which will improve our paper.

1. We moved Aim out from Materials and Methods.

2. Our NICU didn't close to parents during the pandemic, and SSC was always offered to parents.

3. We formatted the tables.

4. We checked table 2, and we confirm it's correct according to the data analysis we performed.

5. We improved the English language.

6. We modified our manuscript in order to avoid long sentences.

Round 2

Reviewer 3 Report

Delete Aim that appeared before the start of materials and methods section. 

Author Response

Dear Reviewer,

thank you for your comment.

We deleted aim from the text.